# A Foundational Population Genetics Investigation of the Sexual Systems of *Solanum* (Solanaceae) in the Australian Monsoon Tropics Suggests Dioecious Taxa May Benefit from Increased Genetic Admixture via Obligate Outcrossing

**DOI:** 10.3390/plants12112200

**Published:** 2023-06-02

**Authors:** Jason T. Cantley, Ingrid E. Jordon-Thaden, Morgan D. Roche, Daniel Hayes, Stephanie Kate, Christopher T. Martine

**Affiliations:** 1Department of Biology, San Francisco State University, San Francisco, CA 94132, USA; 2Biology Department, Bucknell University, Lewisburg, PA 17837, USA; 3Department of Botany, University of Wisconsin Madison, Madison, WI 53706, USA; 4Department of Ecology and Evolutionary Biology, University of Tennessee, Knoxville, TN 37996, USA

**Keywords:** dioecy, fire, population genetics, sexual system, Solanaceae

## Abstract

*Solanum* section *Leptostemonum* is an ideal lineage to test the theoretical framework regarding proposed evolutionary benefits of outcrossing sexual systems in comparison to cosexuality. Theoretically, non-cosexual taxa should support more genetic diversity within populations, experience less inbreeding, and have less genetic structure due to a restricted ability to self-fertilize. However, many confounding factors present challenges for a confident inference that inherent differences in sexual systems influence observed genetic patterns among populations. This study provides a foundational baseline of the population genetics of several species of different sexual systems with the aim of generating hypotheses of any factor—including sexual system—that influences genetic patterns. Importantly, results indicate that dioecious *S. asymmetriphyllum* maintains less genetic structure and greater admixture among populations than cosexual *S. raphiotes* at the same three locations where they co-occur. This suggests that when certain conditions are met, the evolution of dioecy may have proceeded as a means to avoid genetic consequences of self-compatibility and may support hypotheses of benefits gained through differential resource allocation partitioned across sexes. Arguably, the most significant finding of this study is that all taxa are strongly inbred, possibly reflective of a shared response to recent climate shifts, such as the increased frequency and intensity of the region’s fire regime.

## 1. Introduction

During the 19th Century, Charles Darwin commented on challenges in understanding the genetic, demographic, and ecological implications of dioecy in angiosperms. In his seminal work ‘*Forms and Flowers*’, Darwin wrote “*There is much difficulty in understanding why hermaphrodite plants should ever have been rendered dioecious.*” [1]. Dioecy is a sexual system wherein plants of a population are unisexual, with staminate and carpellate flowers on separate individuals. In ‘*Effects of Cross and Self Fertilization in the Vegetable Kingdom*’, Darwin postulated that the selective mechanism driving the evolution of dioecy was a benefit gained through the separation of male and female sexual function thereby alleviating resource allocation costs of both sexes on the same individual, particularly under stressful environmental conditions [2]. Within this framework, Darwin argued against the idea that any advantages gained via obligate outcrossing aiding in inbreeding avoidance were not as significant as improved sexual function due to better resource allocation partitioned across separate plants. This determination was made with the assumption that hermaphroditism (i.e., a sexual system in which individuals of a population exhibit only flowers with both male and female functionality, henceforth referred to in this paper as “cosexuality”) had evolved prior to dioecy in angiosperms. Darwin was only partially correct. Since those initial hypotheses, subsequent studies have shown that both the costs of resource allocation between sexes and outbreeding advantages are important factors shaping the evolution of sexual systems, including dioecy, across different lineages of angiosperms [3].

Despite being a sexual system that can overcome some effects of inbreeding, dioecy is often discussed in terms of being an evolutionary ‘dead-end’ for sexual system evolution because reversions to a sexual system in which male and female function occurs on one individual seemed unlikely, particularly in animals [4]. For angiosperms, the dead-end hypothesis has been supported by the observation that many clades of dioecious taxa are less species rich than their sister taxa that are capable of self-fertilization, suggesting that dioecy might lead to increased extinction rates [5,6] Theoretically, an increased extinction rate or the genetic disadvantages of maintaining a dioecious sexual system could result from a necessity to maintain smaller spatial distributions of individuals in populations since (a) only females contribute to seed production and dispersal of progeny and (b) unisexuality requires the presence of a nearby partner of the opposite sex. Therefore, intrasexual competition for local resources among individuals could, by limiting geographic distributions and local abundance of individuals due to finite resources, have genetic consequences. This would not be the case for cosexual taxa since cosexual flowers exist on all individuals and resources are not in competition between sexes partitioned across separate individuals in different locations. To circumvent a loss of genetic diversity potentially inherent to dioecious sexual systems, dioecious taxa are associated with a number of correlated life history traits whose evolution may be the result of the reallocation of energy that would usually be used by the missing sex into other beneficial traits. Some examples include the evolution of larger fruits with many additional ovules [7] and shifts to wind pollination to bypass issues linked to limited local pollinator density [8,9]. Similarly, woodiness and the often-associated increased longevity of individuals are frequently correlated [7,10,11]. Increased longevity lengthens the duration that optimal combinations of genetic diversity persist in a population, which thereby statistically increases the chances of admixture and introgression of these genes into the future. Similarly, woodiness increases the chances of genetic mixing among populations as it can physically support both larger fruits with more ovules and an increased total number of fruits.

The focal lineage of this study, the “*S. dioicum* + *S. echinatum* Group” [12], is a set of woody *Solanum* taxa of the Australian monsoon tropics (Figure 1) that variously exhibit one of three different sexual systems: cosexuality, andromonoecy (i.e., individual plants with cosexual and staminate flowers borne in each inflorescence), or functional dioecy. Andromonoecious *Solanum* taxa have long been recognized and are most prevalent in the large subgenus *Leptostemonum* [13] of ~550 species [14]. Dioecy was first suggested for the Caribbean species *S. polygamum* Vahl in the 1700s [15], but it was not until the 1970s that additional dioecious *Solanum* taxa were reported [16,17]. As currently understood, dioecy is rare within *Solanum*. Roughly 1% of species, or ≈21 of the ca. 1400 currently described species [18], are dioecious [19,20,21,22,23]. There are an estimated four to six distinct evolutionary events that have led to dioecy in *Solanum* [21,24,25,26,27]. Of these, the largest radiation is from the Australian monsoon tropics, where 13 currently described dioecious species occur [12,23]. Australian taxa may represent either one or two transitions to dioecy and are closely related to a radiation of ca. 15 andromonoecious taxa [28,29] and a clade of ca. 15 cosexual taxa, including the widespread and highly variable *Solanum echinatum* R. Brown [12,29,30] from which *S. raphiotes* is a recent segregate taxon [31]. As is the case with all instances of dioecy in *Solanum* (first formally described by for the Mexican species *S. appendiculatum* Dunal [32]), the condition among the Australian taxa is best described as “functional” dioecy, although these species appear morphologically to be androdioecious [16,17]. The cryptic nature of this system among these Australian taxa manifests itself as female inflorescences which are reduced to a solitary and morphologically appearing cosexual flower bearing anthers producing pollen. However, pollen developed in these anthers is inaperturate and ingerminable [33]. Meanwhile, male plants bear inflorescences that present as a simple monochasial helicoid type cyme consisting of a few to dozens of staminate flowers, all producing porate and germinable pollen [16,17,34]. Additionally, functionally female flowers have larger corollas than those of their male counterparts [33] but produce pollen of lower nutritional quality than the porate pollen produced by staminate flowers [35].

It remains equally likely that dioecy among Australian *Solanum* taxa has arisen from andromonoecious or cosexual ancestry [12,29,33]. Given the rarity of dioecy in the genus *Solanum*, the theoretical work of Martine and Anderson [28] plus Anderson and Symon [33] details necessary transitional steps among *Solanum* sexual systems and the formulation of hypotheses regarding the evolution of dioecy: (a) populations must be small and widely separated from one another, (b) population sizes must be effectively limited due to individual ‘plants’ of a population being the result from only a few ramets of a few surviving genets, and (c) the pollinator fauna must be small and have relatively local ranges such that pollination between widespread populations is infrequent to non-existent. All three of these factors theoretically increase the probability of inbreeding among closely related individuals through self-fertilization. Therefore, these three factors could serve as the selective pressures promoting a physical separation of sexes in order to compensate for the effects of inbreeding and resource allocation. In turn, inducing a mechanism of even partial female or male sterility in a population could play a role in mitigating the effects of inbreeding by allowing for an increase in genetic mixing among andromonoecious or cosexual individuals. Moreover, if outcrossing and sex-related resource reallocation are indeed important to populations of individuals, then dioecy could hypothetically be more efficient in achieving increased genetic mixing among individuals compared to taxa that cannot self-fertilize [36].

No studies have investigated the population genetics of sympatrically occurring Australian *Solanum* taxa directly in the wild. Because sympatric taxa in our study group often have different sexual systems, the group is ideal for testing hypotheses related to the theoretical implications of dioecy in comparison to taxa in which self-fertilization is possible. For dioecy (and perhaps andromonoecy), the aforementioned hypotheses suggest these taxa at some point in time should have quantifiable genetic advantages. Specifically, we hypothesize that patterns observed in population genetics analyses of dioecious species should appear as a greater amount of genetic diversity in populations, exhibit less inbreeding, and be less genetically structured. However, a suite of factors, which are difficult to isolate in situ, present significant challenges for precise inferences on the role sexual systems play in shaping the genetic landscape of taxa. These include microhabitat niche preferences, the duration of site occupation, the degree of initial genetic diversity at the time of population founding events, the typical life spans of each taxon, and the potential for hybridization. Therefore, the primary aim of this investigatory study was to explore population-level genetics of sympatrically occurring *Solanum* taxa occupying the extreme ends of the sexual system variation found in the “*S. dioicum* + *S. echinatum* Group”: dioecy and cosexuality. In doing so, we hope to establish a foundational understanding of how these taxa, operating in the wild, may (or may not) fit into existing theoretical frameworks of sexual system evolution, as well as contribute to the broader understanding of why transitions to dioecy, though rare in number, are still a widespread occurrence among angiosperms.

## 2. Results

### 2.1. Sequencing and Ipyrad Filtering

Across the non-monophyletic assemblage of the five taxa sequenced, a total of 308 variant bi-allelic loci across 193 individuals, representing 10 total populations, were retained from the original 360 original samples sent for sequencing. The removal of data was necessary to produce a ‘hard-filtered’ dataset in which individual samples and loci were retained only when passing the following described strict filtering steps. First, raw sequences were demultiplexed in ipyrad [37], wherein seven samples were removed as they had <200,000 raw sequencing reads. Next, loci were removed if they were not shared across at least 100 of the 358 samples. This resulted in a total of 3288 loci retained. Individuals with >60% missing data were then removed, reducing the number of individuals to 193 and 1458 loci. Following this, 763 loci were removed since they were <10 bp apart from each other. Removal of 50 invariant loci was performed through the application of a 1% Minor Allele Frequency (MAF) threshold. An additional 166 loci were removed as they could be associated with linkage disequilibrium. When filtering for loci that were within 1 kb base pairs of each other and those with an R_2_ value > 0.8, a final 239 linked loci were removed.

### 2.2. Summary Statistics

Summary statistics, geographic locations of sampled populations, and voucher information of each of the five species and all 10 populations are given in Table 1. For species represented by more than one population, species-level statistics are provided as well as statistics for each population. In all cases, regardless of hierarchical level, H_o_ was determined to be significantly less than H_e_. Bartlett’s tests corroborate the statistical significance suggesting that heterozygosity was significantly less than expected at Hardy–Weinberg equilibrium. The F_IS_ values for all five species and 10 populations were all >0.9, indicating that homozygosity is acutely high and suggesting significant levels of inbreeding for all taxa (Table 1).

### 2.3. Pairwise-F_ST_ and AMOVA

Heatmaps of Weir–Cockerham adjusted [38] Pairwise-F_ST_ values were generated at the two hierarchical levels (species, populations) to visualize genetic relationships (Figure 2). When considering species, all Pairwise-F_ST_ values were >0.8, suggesting that the five species are strongly differentiated from each other. At the population level, the three species represented by more than one population (*S. asymmetriphyllum*, *S. raphiotes*, and *S. sejunctum*) each separately have lower Pairwise-F_ST_ values among their own populations, pointing to a moderate to high degree of intraspecific genetic similarity. However, some genetic structure is noted at the population level. Importantly, at the three locations where more than one species was sampled (i.e., Jabiluka, Merl Rock, and Bardedjilidji), Pairwise-F_ST_ values were >0.8, indicating strong genetic structure separates the sympatrically occurring populations of different species with different sexual systems.

Echoing the findings of the Pairwise-F_ST_ values, the results of AMOVA analyses confirm a pattern of genetic diversity with strong species-level structuring and less structure among the populations of species represented by more than one population. Genetic diversity was most pronounced at the species level, where 92.5% of the variance among samples occurs. More variation was found within populations (5.62%) than between populations of a species (1.61%), indicating little genetic diversity among different populations of a species.

### 2.4. PCA and DAPC Analyses

The optimal number of *k*-means clusters useful for describing the data ranged between 8 and 10 as determined by a series of the smallest obtained BIC values. This range parallels the findings of both the AMOVA and Pairwise-F_ST_ statistical analyses. More precisely, the upper limit of *k*-means clusters (10) matches the total number of a priori defined populations, while the lower limit of 8 reflects less genetic structuring among populations of the three species represented by more than one population. The optimal number of principal components (PCs) identified using an *a*-score spline interpolation approach followed by cross-validation was nine. With nine PCs and three discriminant analysis eigenvalues retained, 90% of the variance was represented in this optimally parameterized DAPC. A PCA scatter plot and a compoplot (i.e., a structure-like bar plot) were generated to visualize each individual’s proportional assignment to the 10 genetic clusters (Figure 3). We chose to visualize 10 *k*-means clusters to show the finest scale pattern observed among the populations.

The DAPC structure-like bar plot visualizations (Figure 4) reveal that all five species occupy mutually exclusive *k*-means clusters without a signal of admixture among species, even at the three localities where two species with different sexual systems occur sympatrically. Individuals of two species—*S. ossicruentum* and *S. cowiei*—each represent a unique *k*-means cluster, while the other three species—*S. asymmetriphyllum*, *S. sejunctum*, and *S. raphiotes*—are each represented by two or three *k*-means clusters, reflecting migration among these species’ populations and pointing to a strong shared ancestry. Both *S. asymmetriphyllum* and *S. raphiotes* share a pattern across the three sites where they occur sympatrically. For these two species, Jabiluka reflects a slightly higher level of genetic distinctiveness than Merl Rock and Bardedjilidji localities, which are similar to each other in their proportional assignments of the three *k*-means clusters assigned to each species.

When investigating the genetic structure of populations on a more granular scale using sub-sampled datasets obtained using the repool function of adegenet, a number of additional insights were possible from PCA and DAPC analyses. First, three sub-sampled datasets investigated the within-species relationships of populations. For dioecious *S. asymmetriphyllum* and cosexual *S. raphiotes*, which sympatrically occur at the same three geographic population locations, these finer scale datasets corroborate that both species occupy three *k*-means clusters, with each cluster loosely representing each population locality. However, some patterns of relationships are different between the two species. For *S. raphiotes*, the PCA scatterplot indicates that most variation captured in PC1 represents differences among populations. The distribution of individuals from populations along PC2 is similar, but Merl Rock individuals occupy a wider distribution (Figure 3B). Additionally, Merl Rock has the widest span of the x-axis, suggesting it maintains the most variation of the three populations. *Solanum raphiotes* individuals of Jabiluka and Bardedjilidji do not overlap, but the wider variation of Merl Rock overlaps with both. The DAPC bar plot visualization suggests a signal of gene flow from Jabiluka into Merl Rock and Bardedjilidji populations because of the lack of shared genotypes among the individuals between the two populations (Figure 4B). However, the reciprocal signal of Jabiluka as a sink for gene flow is less prominent as only three individuals are admixed with less than a 0.03 proportional *k*-means cluster assignment from either Merl Rock or Bardedjilidji. *Solanum asymmetriphyllum* differs from *S. raphiotes* in these same locations in several ways. First, the PCA suggests that populations of *S. asymmetriphyllum* are less genetically distinct than they are for *S. raphiotes*, as evidenced by a larger number of admixed individuals (Figure 4C), especially in the population from Bardedjilidji. Unlike the strong genetic structure of the *S. raphiotes* population at Jabiluka, the DAPC of *S. asymmetriphyllum* indicates that Jabiluka is both a stronger source and sink of gene flow. There is also greater exchange between Merl Rock and Bardedjilidji. Overall, (i) the total number of admixed individuals, (ii) an overall greater proportional assignment of the Jabiluka *k*-means cluster to individuals of *S. asymmetriphyllum* at Merl Rock and Bardedjilidji, and (iii) the observation that Jabiluka is both a more significant source and sink of genetic diversity suggests that migration between populations of *S. asymmetriphyllum* is more readily achieved and subsequently maintained than it is for *S. raphiotes*.

*Solanum sejunctum* is represented by only two populations, Gubara Pools and Barkk Sandstone. In the DAPC analysis, each population is proportionally represented, predominantly by their own unique *k*-means cluster; however, each maintains a weak signal of admixture from the other population (Figure 4C). Nearly half of the individuals from each population are admixed. One individual in each population has a greater than 0.50 proportional assignment of the other population’s *k*-means cluster, but all other admixed individuals have proportional assignments of the other population’s *k*-means cluster at levels less than 0.25. When assessing the placement of individuals in the scatterplot of PC1 vs. PC2, individuals of populations overlap partially; however, there is obvious clustering by a priori population assignment. Generally speaking, *S. sejunctum* exhibits a similar pattern to *S. asymmetriphyllum* and *S. raphiotes* in terms of the number of admixed individuals (Figure 4D).

Side-by-side comparisons of species pairs that occur sympatrically in PCA and DAPC analyses yield interesting insights (Figure 5). First, no admixture is observed between either *S. asymmetriphyllum* or *S. raphiotes*. In both repooled datasets, PCA analyses separate species along the x-axis of PC1, indicating this is where the majority of variation is located (between species). Variation along the y-axis of PC2 indicates the variation within individuals, and populations of the species echo the patterns observed in the species-specific repooled datasets. Overall, cosexual *S. raphiotes* occupies more variation than dioecious *S. asymmetriphyllum*.

### 2.5. Isolation by Distance

Mantel tests were performed for *S. asymmetriphyllum* and *S. raphiotes* to test for an isolation by distance (IBD) model. Mantel tests for both taxa indicated that the genetic structure of these populations does not follow a classic IBD model. Simulated *p*-values of 0.3334967 and 0.5000145 were recovered for *S. asymmetriphyllum* and *S. raphiotes*, respectively. Local density measured using a 2-dimensional kernel density estimation reveals that Jabiluka represents a ‘distant patch’ rather than clinal variation, as would be expected in a classic IBD model. One difference is noted between the populations of the two species. The variation in *S. asymmetriphyllum* at Bardedjilidji and Merl Rock appears to be one genetic cluster that is clinal, but *S. raphiotes* retains greater genetic separation among these localities.

## 3. Discussion

### 3.1. Genetic Diversity and Structure in Dioecious Australian Solanum

Our findings offer support for both the avoidance of inbreeding and resource allocation hypotheses for the selective pressures promoting the evolution of dioecy in Australian *Solanum*. However, the patterns of population structure and diversity recorded for four different dioecious taxa suggest (i) a pattern for *S. asymmetriphyllum* with reduced variation compared to that of the sympatrically occurring cosexual taxon *S. raphiotes* (though with widespread admixture) and (ii) two dioecious taxa, *S. ossicruentum* and *S. cowiei*, with limited variation. Patterns inferred for *S. asymmetriphyllum* and potentially for *S. sejunctum* leave open the possibility that dioecy may have evolved as a mechanism to avoid the genetic cost of inbreeding in dioecious Australian *Solanum* taxa. A direct comparison of the genetic structure of dioecious *S. asymmetriphyllum* and cosexual *S. raphiotes* at three co-occurring locations (Jabiluka, Merl Rock, and Bardedjilidji) reveals that migration between populations of *S. asymmetriphyllum* is more readily achieved (thus genetic diversity is decreased) and subsequently maintained than it is currently for *S. raphiotes* (which shows higher genetic diversity). All analyses conveyed a similar finding; across three localities, *S. asymmetriphyllum* and *S. raphiotes* were similar in overall variance and levels of inbreeding, but *S. asymmetriphyllum* importantly maintains more admixture across all populations. This pattern suggests that symmetrical migration among populations of *S. asymmetriphyllum* has occurred and was subsequently maintained. We can therefore conclude that the gene pools of this dioecious taxon are continuously being mixed, resulting in decreasing overall genetic diversity. Comparatively, *S. raphiotes* maintains less admixture at the same population locations, and the pattern of migration is asymmetrical. The gene pools of this cosexual taxon indicate that the presence of balanced male-to-female floral parts requires less migration between populations to maintain the species. This, therefore, results in higher genetic diversity between populations as fewer genes are shared over time. Specifically, the *S. raphiotes* population at Jabiluka appears to currently be only a source of introgressive gene flow to Merl Rock and Bardedjilidji. Given that H_o_ is significantly lower than expected and F_IS_ values are elevated to similar levels across both taxa, it is possible that the admixture observed among *S. asymmetriphyllum* populations represents a beneficial mechanism of dioecy that helps avoid genetic consequences of self-fertilization through introgression via obligate outcrossing and maintenance of interspecific hybrids from nearby populations. Interestingly, the populations of both *S. asymmetriphyllum* and *S. raphiotes* found at Bardedjilidji seem to have the most diverse genetic makeup; perhaps this is a function of sharing the same local suite of pollinators and/or seed dispersers, but we cannot be certain. However, an important caveat to the conclusions drawn above is that one must assume demographic history at the site is similar for both species to allow for these comparisons. Demographic differences, such as time of site occupation (i.e., founder events) or recent bottleneck events, and even ecological niche preferences could be influential factors driving the different patterns observed between these two species.

While fundamental data regarding certain life history traits and site demographics remain unknown and beyond the scope of this study, novel insights help sharpen the evolutionary understanding of selective pressures that may have shaped the genetic landscape of dioecious vs. cosexual taxa of the Australian monsoon tropics. Several known correlates of dioecy are differentially expressed in these taxa, including pronounced secondary growth resulting in larger and taller individuals, larger fruits each with hundreds of seeds, larger female corollas, and more conspicuous male floral displays [34,39]. Additional factors shaping population structure and gene flow among populations—while not known to be necessarily correlated with dioecy—may also be important, such as inflorescences that are elevated higher above the ground, greater fecundity via an increased number of fruits per individual per season, and the potential exclusion of smaller vertebrate dispersal vectors resulting from larger and heavier fruit and seed sizes. There are also abiotic factors, such as soil chemistry, fire, and precipitation regimes (i.e., abiotic environmental stochasticity), spatiotemporal patch dynamics, and the finite availability of suitable habitat. However, our clearest inference detailing the evolutionary strategy for dioecy in the Australian monsoon tropics is for three Kakadu dioecious *Solanum* (KDS): *S. asymmetriphyllum*, *S. sejunctum*, and the southern undescribed taxon *S.* ‘sp. Deaf Adder’ [40]. KDS taxa are each other’s closest relatives [12,30] and occupy very similar ecological niches. All have the intriguing ability to produce clonal genets arising from underground rhizomes emerging through cracks and fissures of dissected sandstone [34,41]. Underground rhizomes of KDS taxa are not unique for Australian *Solanum* [42,43], but it seems KDS and closely related dioecious taxa are more strongly reliant upon this ability to regenerate post fire, hinting at a strategy for long-term site occupancy in the finite locations of suitable dissected sandstone habitat. Although the average longevity of KDS genets is not known, in situ observations and greenhouse experiments suggest they likely persist many generations longer than most cosexual Australian *Solanum* taxa of the focal lineage [34]. An additional factor of consideration regarding increased longevity as part of the evolutionary strategy of KDS taxa must also include the formation of a genetically diverse soil seed bank. Fruit-producing KDS plants annually produce thousands, if not tens of thousands, of seeds that lay dormant in the soil until optimal conditions are met for germination [33,34]. The genetic structure of this seed bank theoretically represents half maternal genetics, and when the mother plant eventually senesces, perhaps following fire, competition among seedlings should allow for its replacement by the most optimal combination of beneficial alleles stored in the bank. Consideration of these three factors—increased longevity, prolific post-fire resprouting, and comparatively enormous seed banks—may have evolved as a result of newly gained flexibility in resource allocation correlated with the evolution of dioecy aiding in increased reproductive assurance and may explain the pattern of maintained admixture observed in our data.

In contrast to the wider variation observed for dioecious *S. asymmetriphyllum* and *S. sejunctum*, populations of dioecious *S. cowiei* and *S. ossicruentum* appear to be less genetically diverse (despite the lack of direct comparisons with sympatric cosexual taxa), though similar patterns of low heterozygosity and high F_IS_ values were found. Several factors could explain this contrasting pattern. Both taxa may still fundamentally represent the same evolutionary strategy as KDS taxa as they are noted to have many of the same adaptations, such as large fruits with hundreds of seeds, underground rhizomes allowing for post-fire resprouting, and large seed banks. Some notable differences among these taxa are simply the locations that they inhabit—both *S. cowiei* and *S. ossicruentum* occur as many fragmented populations across a much wider and more western distribution [20,22], which may have had different stochastic environmental conditions. In particular, both the precipitation and fire regimes of these areas could have differentially caused bottlenecks or led to recent founding events different from those of KDS taxa. Furthermore, *S. cowiei* and *S. ossicruentum* may be descendants from an independent evolution of dioecy due to differential life history traits compared to the KDS sandstone endemic taxa, thus allowing them to evolve along divergent paths and ultimately resulting in the reduced variance observed.

From this study, it seems that the genetic status of dioecious Australian *Solanum* could support both the avoidance of inbreeding and resource allocation hypotheses for the selective pressures promoting the evolution of dioecy; however, more research is needed. These taxa are an ideal system in which to investigate how variance among correlated dioecious life history traits variously contributes to resilience to environmental stochasticity. For example, some traits of *S. cowiei* and *S. ossicruentum* are not as pronounced and may have had less of an evolutionary benefit for maintaining genetic diversity when compared to KDS taxa. Obvious differences include smaller, dry, bony fruits with tiny seeds in *S. ossicruentum* [22] and the growth habit of *S. cowiei* that includes the production of rhizomes directly in sand at the base of sandstone boulders, which may offer less, or at least differential, protection from fire and herbivory [20]. In short, it may be that differences across a suite of life history traits, demographics, and environmental stochasticity of a heterogeneous landscape may have led to the wider variation (and assumedly greater evolutionary fitness) observed for KDS taxa.

### 3.2. Insights from a Shared Background of Elevated Inbreeding

Some of the most striking results of this study are the ubiquitously elevated levels of homozygosity and the high values of the inbreeding co-efficient (F_IS_) across all taxa. This indicates that inbreeding is an important selective pressure for all taxa. It is suggested that heterozygotes should be selected for over homozygotes on account of ‘hybrid-vigor’, which should benefit an individual’s evolutionary fitness in comparison to homozygotes. Moreover, because homozygotes are thought to perpetuate deleterious alleles in a population, homozygotes are subject to the effects of inbreeding depression [44]. Evidence for the selective favoring of heterozygotes is supported in several studies through the demonstration of differential heterozygosity among plants of different size classes or generations [44,45], though no such pattern is observed here for *Solanum*. An obligately outcrossing sexual system of dioecy should theoretically mitigate the effects of inbreeding (at least at the time of its evolution), but mitigation can only persist in large populations of many individuals that allow for outcrossing to occur among more distantly related individuals rather than outcrossing with siblings, which would subsequently increase homozygosity in the population.

Our results raise the question as to why inbreeding and homozygosity are so pronounced in our data. Our data only provide a snapshot in time of the life histories of these *Solanum* taxa. These species differ from each other in several ways (sexual system, longevity, fecundity, size, specific ecological niche preference, etc.), but all have evolved and adapted to the climatic processes of the Australian monsoon tropics [46]. The AMT is a dynamic biome that has had a fluctuating climate for many millions of years. With the Pliocene closure of the Isthmus of Panama 3.0–2.5 Ma [47,48], the modern circulation patterns of Earth’s oceans were established and several cycles of glaciation–interglaciation followed. During warmer interglacial periods, Australia became more arid overall, though in the north the monsoon cycles strengthened, bringing more precipitation to the region. This would have resulted in the production of more biofuel, effectively changing the intensity and frequency of fires. One product of this was the emergence of fires as a principal ecological process across the AMT [49,50]. However, the once much more contiguous rainforests of the region contracted as the overall aridification of the continent commenced, resulting in intensified drought periods following periodic monsoonal precipitation events. Aridity in the Kimberley region was more pronounced, and it is suggested that the lower biodiversity observed there today is in part a result of a higher extinction rate driven by these intensified droughts in comparison to the sandstone flora of the Top End in the Northern Territory [46,51]. During periods of glaciation, global temperatures were cooler, resulting in weakened monsoon cycles with less precipitation and less accumulation of organic material to fuel fires. As a result, fires may not have been a significant abiotic pressure shaping the ecosystems during these glacial periods [46].

AMT fires have been characterized as likely to have been infrequent and of higher intensity prior to human arrival [49,52,53]. These landscapes were host to far fewer fire-adapted species, a protracted savanna of fire-promoting trees and grasses, widespread evergreen dry forests, and more contiguous rainforest habitat. All *Solanum* taxa of this study are associated with patchily distributed sandstone escarpment habitats existing throughout a widespread and fire-prone savanna of open eucalypt woodland on plains of oligotrophic sandy soils [22,34]. This heterogenous matrix of habitats influenced by stochastic processes of fire provides context into the finding that populations of *S. asymmetriphyllum* and *S. raphiotes* were not found to be isolated by distance. Many taxa of the sandstone flora have narrow ranges of specific habitat restricted to deep gorges, steep escarpments, and cliffs [54]. Today, sandstone habitats harboring *Solanum* taxa range from *Allosyncarpia* (Myrtaceae)-dominated rainforest relic patches to xeric heathlands—habitats that are partially protected from fires due to the topographic complexity of their terrain. As with many endemic sandstone taxa, AMT *Solanum* are thought to have evolved strategies for survival with fire. For example, although KDS taxa can resprout following fire via underground rhizomes or germinate post fire from large seedbanks, they also are largely restricted to these partially fire-protected refugia.

The arrival of the first humans to the AMT ushered in a regime of frequent, small, and low-intensity fires that likely contributed to an expansion of fire-tolerant and fire-adapted species coincident with the retraction of fire-sensitive taxa to habitat refugia [55,56,57]. Much more recently, human alterations to the environment have initiated a stark shift in the regional fire regime [58], which is likely a major and alarming factor in the abundant homozygosity and inbreeding we have documented in this study. Beginning with the arrival of Europeans less than 200 years ago, an unambiguously different fire regime now dominates the Top End, characterized by fires that are more extensive, frequent, and intense than occurred previously under the traditional management of indigenous peoples [54,55,56,57]. Consequently, the AMT has seen a recent expansion of grassy understories. In turn, this produces more flammable biomass, promoting more intense and frequent fires [59]. Throughout this rapid shift in regional fire regime (<200 years), fire-tolerant *Solanum* taxa sensitive to higher fire temperatures at longer durations would become more and more restricted to many fragmented refugial habitats and forced to primarily exchange genes with small numbers of individuals in close proximity, increasing the possibility of newly elevated levels of inbreeding.

The extreme inbreeding and homozygosity observed in these Australian *Solanum* taxa fits with previous research indicating that anthropogenically driven fire regime changes over the last two centuries are directly responsible for significant biodiversity loss and a contraction of suitable habitat for many taxa across sandstone environments [54,60,61,62]. The two habitats with the highest endemism—heathlands and monsoon rainforests—are documented as having ongoing decline in the abundance of fire-sensitive taxa [56,63,64,65]. Research indicates that over 40% of sandstone vegetation in Kakadu National Park alone had been burnt over a 14-year period from 1980 to 1994 at frequencies of at least one year in every three years [54]. This work found the risk of extirpation due to fire was particularly acute for obligately reseeding species such as *Solanum*, which comprise over half of the heathland flora sampled, thus highlighting these taxa as a conservation management priority.

## 4. Materials and Methods

### 4.1. Taxon Sampling and Field-Observed Life History

Initial collections for this study included samples from 360 individuals representing five *Solanum* species across three sexual systems from 10 locations of the Top End region of the Northern Territory, Australia, during three collecting expeditions from 2009 to 2014 (Table 1, Figure 1). Ultimately, the only samples retained following strict filtering steps required for downstream analyses (see below) were those of taxa representing the sexual systems of primary interest in this study: cosexuality and dioecy.

All taxa in this study share a common Australian ancestor [30,66], estimated at approximately 5 million years old [67]; chosen taxa represent a polyphyletic assemblage. Four taxa have a dioecious sexual system (*S. asymmetriphyllum* R.L. Specht, *S. cowiei* Martine, *S. ossicruentum* Martine and J. Cantley, and *S. sejunctum* K. Brennan, Martine, Symon), and *S. raphiotes* A.R. Bean is cosexual [31]. The knowledge of relationships among this group of Australian *Solanum* taxa is still in flux, but of the four dioecious taxa, *S. asymmetriphyllum* and *S. sejunctum* are sister taxa (i.e. Kakadu Clade) [66] and are in turn sister to a clade of cosexual taxa, which includes *S. raphiotes* [12]. These two clades appear to be reciprocally sister to a larger dioecious clade of 13 taxa (i.e., Kimberley Dioecious Clade) [23,66], which include dioecious *S. cowiei* and *S. ossicruentum*. These dioecious and cosexual taxa are in turn sister to a clade of ca. 15 andromonoecious taxa (i.e., Andromonoecious Bush Tomato Clade) [66]. Evidence of hybridization among any of the taxa with different sexual systems has not been observed in the field via intermediate morphological characters, but attempts to form hybrids of taxa with different sexual systems in a greenhouse setting have proven occasionally successful [68] and are currently under study [69].

The sampling strategy for the collection of populations included three sites where two *Solanum* species with different sexual systems were present. We were able to collect three population pairs of dioecious *S. asymmetriphyllum* and cosexual *S. raphiotes* at three locations in Kakadu National Park where they sympatrically occur: at Merl Rock, Bardedjilidji, and Jabiluka. During collection, populations were discerned by geographic limitations of localities where collections could be safely made. Notably, individuals of all taxa, except for *S. raphiotes*, tended to occur in higher densities around sandstone escarpment and boulders. This differential in density may play a role in the genetic patterns observed in our data. However, in an effort to mitigate this potential effect, we collected individuals of both cosexual and dioecious taxa occurring as evenly as possible across the same accessible geographic area.

### 4.2. DNA Library Preparation and Sequencing

Genomic DNA was extracted from dried leaf material using a modified CTAB method [70]. Leaf tissue was pulverized using a GenoGrinder (SPEX Sample Prep, Metuchen, NJ, USA) and steel beads in 2 mL microfuge tubes. Leaf material was hydrated in lysis buffer and incubated for 30 min at 37 °C and centrifuged, and the supernatant was then extracted with a chloroform:isoamyl alcohol (24:1) solution before being precipitated at −20 °C in ice-cold 100% isopropanol for 20 min. DNA pellets were cleaned with two consecutive washes of ethanol, 75% and 95%, respectively, resuspended in ddH_2_O, and then incubated at 37 °C with RNase A (Sigma-Aldrich, St. Louis, MO, USA) for one hour. For restriction enzyme digestion, we found that samples benefited from desalting, which resulted in more complete and clean digestion in preparation for double digest restriction enzyme site-associated DNA sequencing (ddRADseq) library preparation following a slightly modified protocol [71,72]. Desalting was conducted by repeating the above isopropanol precipitation step and pellet washes with ethanol. DNA concentration was tested with a Qubit 2.0 (Invitrogen, Waltham, MA, USA) using the High Sensitivity Kit, and the samples were then normalized in concentration to have ≈300 ng of DNA per sample in a 96-well plate with a concentration of 60 ng/μL, adding one well of a purchased commercial control gDNA of *Solanum tuberosum* (BioChain, Newark, CA, USA) in each plate. EcoRI HF and SphI HF restriction enzymes were chosen bioinformatically for gDNA digestion as they have comparable cut site frequency for the focal *Solanum* taxa, as determined in silico using *Solanum tuberosum* as a closely related reference genome using Geneious (San Diego, CA, USA). All adapters, primers, and barcodes were ordered from Eurofins Operon (Louisville, KY, USA). All enzymes used within (T4 DNA Ligase, Phusion PCR kit, EcoRI HF, SphI HF) were ordered from NE Biolabs (Ipswich, MA, USA).

Using 300 ng of gDNA template at 60 ng/μL (5 μL volume), 0.75 μL of SphI-HF and 0.75 μL of EcoRI-HF (15 units each per reaction), 2.5 μL of 10× CutSmart Buffer, and 16 μL of water (making a total of 25 μL of reaction volume per well), the DNA was digested for 3 h at 37 °C in an incubator, kept at 4 °C, and then cleaned using AMPure XP Beads as suggested by the manufacturer (Beckman Coulter). This and the following procedural tests were performed to determine the optimum number of digestion enzyme units (5 to 25 units) with a sample pool of 48 individuals that were not the experimental samples but instead from extra spiny *Solanum* leaf tissue collected from greenhouse grown plants. This digested and cleaned DNA was quantified with a Qubit BR Kit and normalized from 0.1 to 0.2 μg (100 to 200 ng). The annealed/barcoded adapters were ligated to the digested and cleaned DNA samples using a simple mixture of adapters, 2 μL of each primer, 2 μL of T4 DNA Ligase, and 4 μL 10 X ligase buffer per well added to 30 μL of digested DNA, resulting in a total ligase reaction volume of 40 μL per sample. The selected barcodes and adapters were ligated in a thermocycler according to the following routine: heating to 37 °C for 30 min, heating to 65 °C for 10 min, cooling slowly by 2 °C every 90 sec until 21 °C was reached, holding the temperature at 21 °C for 10 min, holding the temperature at 4 °C, and then cleaning using AMPure XP Beads. Digested DNA fragments with ligated adapters and barcodes were then size-selected using a 10 mm thick 2% agarose maxi-gel and prestained with 17.5 μL of GelRed (Phenix Research, Swedesboro, NJ, USA), with 2 combs with a thickness of 1.5 mm and 20 wells; the pools of 48 barcoded samples each were loaded into the wells after being mixed with 10 μL of 10× GoTaq Green buffer (undiluted) (Promega, Madison, WI, USA). Using the UV lamp on the gel visualization box (Major Science, San Jose, CA, USA), the gel regions between 200 and 500 bp were cut from the gel with a razor blade and frozen at −20 °C before then being placed in a 2 mL cellulose filter tube (Spin X Column, Costar, Washington, DC, USA) and centrifuged at room temperature for 30 min at 13,000 rpm. To ensure all DNA was removed from the agarose, 100 μL of water was added, and the columns were spun again for 10 min at 15,000 rpm. Each pool was cleaned with the AMPure XP beads again with 1.5 X beads to pool volume to reach a final volume of ~30 μL and quantified with either the Qubit BR or HS Kit to ensure 100 to 300 ng of DNA per pool. For the final library PCR amplification step, each pool was normalized to 100 ng of DNA. The PCR reaction used the Phusion High Fidelity PCR Kit (200 reactions per kit) and included 8.5 μL of water, 4 μL of 5 X HF Buffer, 0.4 μL of 10 μM dNTPs, 0.6 μL of DMSO, 2.0 μL of 10 μM PCR1 primer, 2.0 μL of 10 μM PCR2-index primer (1 to 12 indices), 20–25 ng of DNA/water in 10 μL, and 1.0 μL of DNA Phusion Polymerase per reaction. The thermocycler settings were as follows: step 1: 98 °C for 30 s; step 2: 98 °C for 10 s; step 3: 79 °C for 30 s; step 4: 72 °C for 30 s; step 5: go to 2, 14 times; step 6: 72 °C for 10 min; step 7: 4 °C forever. Tests were performed to determine the best number of cycle repetitions (from 5 to 30) for optimum library amplification. These tests were conducted on a sample pool separate from the experimental sample set. To determine the success and quantity of the pooled library amplifications, the concentration of the libraries before and after PCR, as well as before and after magnetic bead cleaning of the libraries, were determined with the Qubit HS Kit. Using this method, we could determine that optimum and/or successful amplification was on average 50 times the original product. Gels of amplified libraries were not used as it was found that too much material was lost and visualization was inconsistent. Each final amplified library was quantified with a Bioanalyzer (Agilent, Santa Clara, CA, USA) fragment analyzer at iBEST (University of Idaho), which averaged ~37 ng/μL per pool, ranging from 1000 to 6000 ng of total amplified DNA. In some cases, the pools were size-selected again with PippenPrep at iBEST to remove small molecular weight contaminants before sequencing. Each pool was sequenced at the QB3 Vincent J. Coates Genomics Sequencing Laboratory at the University of California Berkeley on an Illumina MiSeq in a 150-paired-end read rapid run. Four pools of 48 barcoded and indexed samples were combined into one lane, resulting in 192 individual samples per lane (384 individuals were sequenced in two lanes).

### 4.3. Sequenced Data Processing and ‘Hard-Filtering’

Raw paired-end Illumina sequencing reads were demultiplexed, followed by the removal of their adaptors and de novo assembly into loci using the program 3.0.63 ipyrad [37]. All Fastq sequence files are accessible from GenBank’s National Center for Biotechnology Information Short Read Archive database (SUB4712600/PRJNA498556). Two samples were removed prior to assembly, which had low numbers of sequencing reads. Loci were removed if they were not shared across at least 100 of the 358 samples. Using the output from the ipyrad run, vcftools [73] was used to identify individuals with >60% missing data, which were then systematically removed. Loci with >50% missingness and loci that were very close to each other (<10 bp) were then removed. Output files were converted to formats compatible with plink [74], where additional filtering of minor alleles with frequencies of >1% plus linked loci within 1 kb of each other and with R_2_ values > 0.8 to account for linkage disequilibrium were removed. The resulting ‘hard-filtered’ dataset reduced the total number of individuals to 193, representing 1458 loci. A full detailing of all parameters used in ipyrad assembly followed by hard filtering performed in vcftools and plink are available in Appendix A, respectively.

All following analyses were conducted using R Studio (ver 1.4.1717, Rstudio, Boston, MA, USA) and associated packages [75]. Descriptive statistics such as F_ST_, F_IS_, and observed and expected heterozygosity (H_O_, H_E_) were calculated using the packages hierfstat [76], pegas [77], and adegenet [78]. Pairwise-F_ST_ values were calculated at two different hierarchical levels: among the 10 populations and separately for the five species. Both calculations implemented the Weir–Cockerham correction method of Pairwise-F_ST_ estimation to account for differences in the number of individuals of each species or population [38]. Bartlett’s tests were conducted at both hierarchical levels to investigate the statistical significance of variance among calculated values of H_O_ and H_E_. To further understand the variance at the different hierarchical levels, an AMOVA was performed in the package poppr [79].

The genetic structure of the populations and species was estimated and visualized using two multivariate methods: Principal Components Analysis (PCA) and Discriminant Analysis of Principle Components (DAPC) on the hard-filtered dataset in the adegenet program. Prior to running PCAs, we first used *k*-means clustering to identify the appropriate number of genetic clusters as indicated by the lowest Bayesian information criterion value using the find.clusters function. Principle Component (PC) data were then transformed by discriminant analysis. DAPC requires users to define the number of PCs retained in the analysis to help mitigate the generation of questionable results since too few or too many retained PCs can affect the balance between overfitting the data and the power of the discriminant analysis. We analyzed between 3 and 250 PCs and then used an *a*-score spline interpolation approach (i.e., the proportion of successful reassignments of group membership of all individuals in the dataset corrected for the number of retained PCs) followed by implementation of the xvalDAPC function for cross-validation to determine that 9 PCs was the optimal number to retain before discriminant analysis. To visualize the DAPC, we generated a ‘compoplot’ (i.e., a structure-like bar plot) in which each individual was proportionally assigned to one or many *k*-means clusters to illustrate population and species membership probability and to identify admixed individuals.

To understand different axes of variation in our dataset for species and populations at a more nuanced level, the hard-filtered dataset was sub-sampled in several ways using the repool function before additional PCA and DAPCs were performed. The four sub-sampled datasets were (i) the two populations of *S. sejunctum*, (ii) three populations of *S. asymmetriphyllum*, (iii) three populations of *S. raphiotes* and, (iv) the combination of dioecious *S. asymmetriphyllum* and cosexual *S. raphiotes* used to compare the variation in sympatrically occurring species with different sexual systems at three different geographic locations (Jabiluka, Merl Rock, and Bardedjilidji).

For *S. asymmetriphyllum* and *S. raphiotes,* which each had three populations, we used Mantel tests to estimate whether isolation by distance influenced the observed distribution of genetic diversity of these taxa. Additionally, as isolation may not be the only correlating factor of geographic distance to genetic distance, we explored the subtle difference of whether populations of each species conformed more to predictions of clinal genetic differentiation (i.e., a classic IBD scenario) or if they functioned more similarly to a model of distant patches in which each population is genetically differentiated and distantly located but not behaving in a cline-like fashion. This was explored by plotting an IBD plot using the 2-dimensional kernel density estimation function kde2d in adegenet.

## 5. Conclusions

Arguably, our most significant finding is that all five study species of Australian *Solanum* of the Australian monsoon tropics are highly inbred. Secondarily, and more directly addressing the main hypotheses tested in this research, no pattern of genetic structure or diversity can be definitively attributed as resulting from the presence of a sexual system alone. However, in the portion of the complete dataset that we were able to compile, we can confidently say that the dioecious taxa have decreased genetic diversity and more admixture between populations in comparison to the cosexual taxon. Several traits present in the focal taxa—which may or may not be correlated to dioecy—likely shape the genetic landscape of each taxon, but conclusions could not be made given the limited populations that ended up in the final cleaned sequence dataset. When all factors are taken into consideration, our data suggest differential combinations of sexual system, correlated life history traits, and demographic history of populations better explain the observed patterns than sexual system alone. Furthermore, when certain conditions are met, obligately outcrossing dioecious taxa may be more capable of maintaining a greater degree of admixture among populations than cosexual taxa. These insights complement the theoretical framework hypothesizing that the evolution of dioecy in Australian *Solanum* may have proceeded as a means to avoid the genetic consequences of self-fertilization and may even support hypotheses of benefits gained through differential resource allocation partitioned across male and female individuals. Emerging from our data are new hypotheses of a testable multi-factorial framework in which benefits of the evolution of dioecy can be teased apart for a more nuanced understanding in future research.

An alarming finding of this research was elevated homozygosity across all taxa, regardless of the sexual system. Although it appears that *Solanum* taxa have some evolved strategies to live in the presence of fire, many of these taxa are presumably fire-sensitive, and the recent human-mediated shift in regional fire regime looms large as a significant factor fueling reduced genetic diversity across many taxa of the Australian monsoon tropics. Frequent fires are noted as occurring more often than the time necessary for obligate reseeding taxa such as *Solanum* to reach full reproductive maturity. From a standpoint of genetic diversity and inbreeding, these selective forces could certainly promote a high occurrence of inbreeding for taxa such as *Solanum* that are tolerant of historically infrequent low-temperature fires but inadequately adapted for recent and future fire intensities. In fact, future studies should be designed to assess the hypothesis that high levels of inbreeding and elevated homozygosity may be a common feature for many fire-sensitive sandstone taxa of the Top End, and perhaps across all of the Australian monsoon tropics. Resultantly, this work highlights a pressing need to include fire-sensitive taxa in the AMT as an important conservation priority for conservation managers.

## Figures and Tables

**Figure 1 plants-12-02200-f001:**
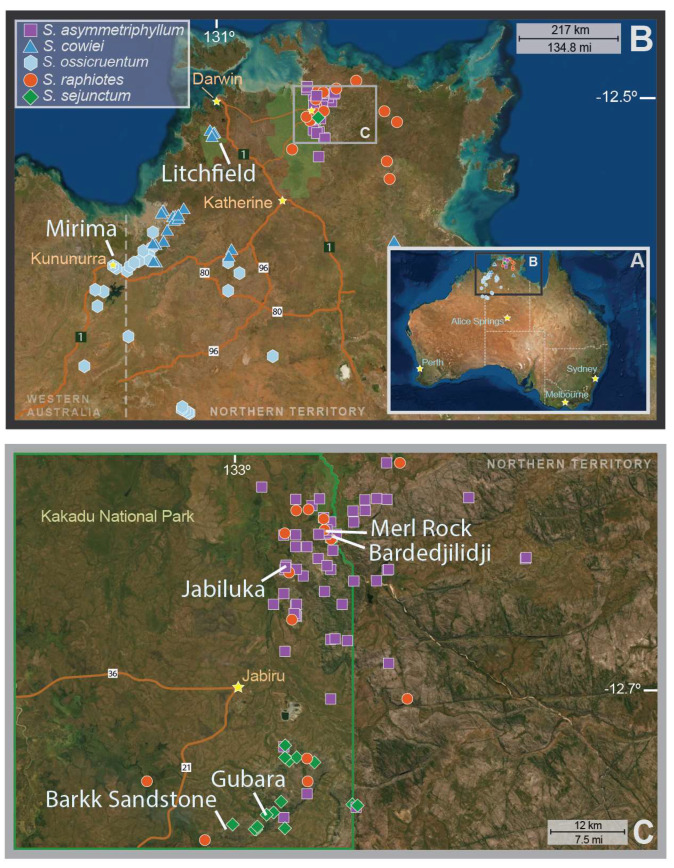
Map indicating known herbarium collection localities of each of the five species for which population genetics data were analyzed in this study. A view of collections within (**A**) Australia, (**B**) the northwest region, and (**C**) a closer view of Kakadu National Park and the Arnhemland Plateau. White lines and text indicate population collection localities used in this study (note that Merl Rock and Bardedjilidji populations are discrete and separated by approximately 2.5 km despite overlapping collection icons). Stars indicate human population centers along roads (denoted by orange lines). Green polygons in (**B**) indicate Litchfield National Park and Kakadu National Park.

**Figure 2 plants-12-02200-f002:**
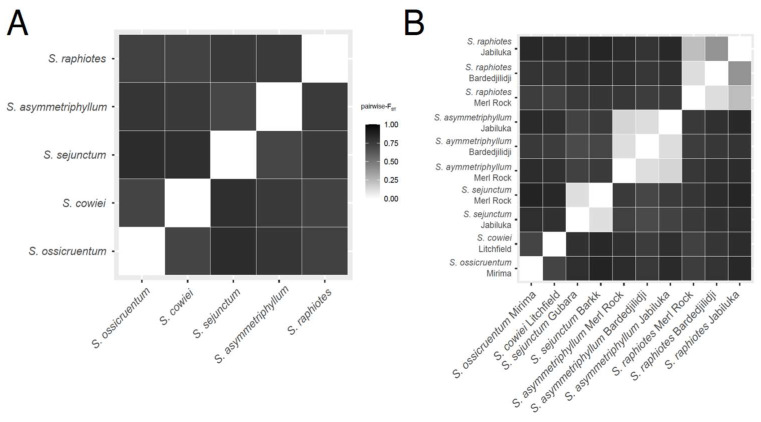
Heatmap of Weir–Cockerham-adjusted Pairwise-F_ST_ values among (**A**) the five Australian *Solanum* taxa of this study and (**B**) among all 10 sampled populations from 193 total individuals and 308 bi-allelic single-nucleotide polymorphisms.

**Figure 3 plants-12-02200-f003:**
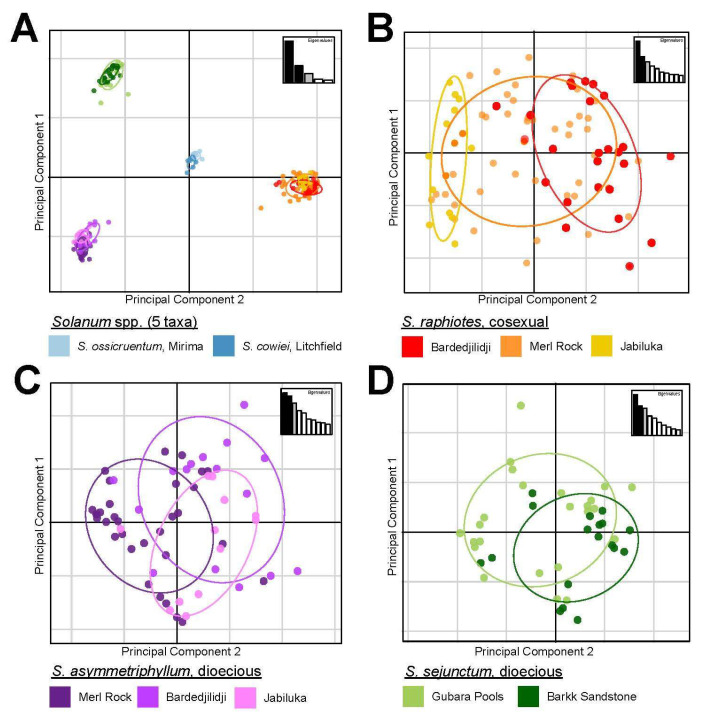
Multivariate analysis visualizations of all five species representing all 10 genetically determined populations of this study from 212 total individuals and 308 bi-allelic single-nucleotide polymorphisms. (**A**) A PCA scatterplot comparison of PC1 and PC2 for all five taxa in this study. Colored points and their corresponding labels indicate species and population. (**B**) PCA for *S. raphiotes*. (**C**) PCA for *S. asymmetriphyllum*. (**D**) PCA for *S. sejunctum*.

**Figure 4 plants-12-02200-f004:**
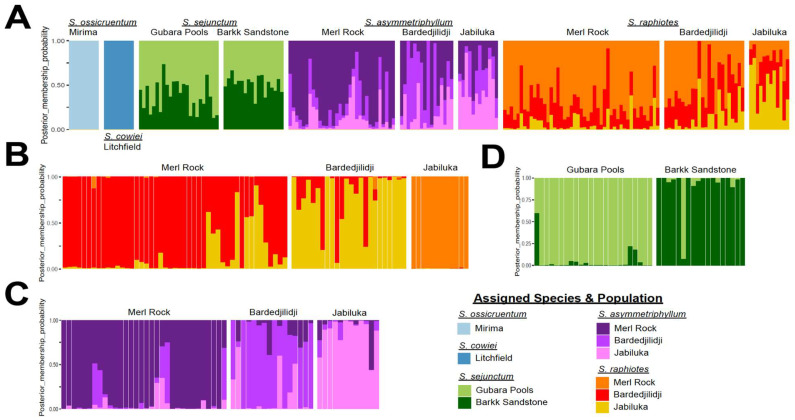
Results of multivariate DAPC analyses for all species (**A**) and sub-sampled analyses of the three individual species that are represented by more than one population (**B**–**D**). Each panel visualizes a *k*-means DAPC structure-like bar plot. This visualization of the DAPC retains the first three eigenvalues and the optimal nine PCs. Vertical bars represent the proportional assignment of each individual to one or several of the eleven *k*-means clusters, as indicated by colors in the legend. Species and populations are as indicated along the x-axis.

**Figure 5 plants-12-02200-f005:**
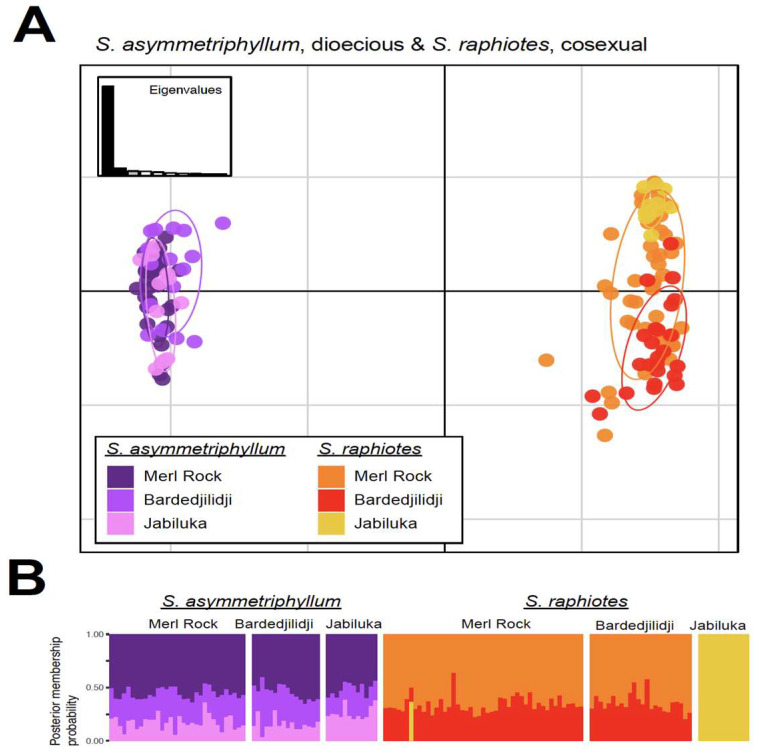
Visualizations of multivariate analyses of the three locations with sympatrically occurring *Solanum* taxa with different sexual systems. The top panel (**A**) is a PCA scatterplot and below (**B**) is the DAPC structure-like bar plot for dioecious *S. asymmetriphyllum* and cosexual *S. raphiotes*, each from Merl Rock, Bardedjilidji, and Jabiluka.

**Table 1 plants-12-02200-t001:** The sexual system, geographic location, reference vouchers, and summary statistics of the five species and 10 populations of this study.

Species	Sexual System	Population	*n*	Latitude	Longitude	Reference Voucher (Herbarium)	H_o_	H_e_	F_IS_
*S. ossicruentum*	Dioecious	Mirima	9	15.76378	128.75175	CTM 4011 (BUPL)	0.0014	0.0924	0.9849
*S. cowiei*	Dioecious	Florence Falls	9	13.21958	130.73645	CTM 1751 (BUPL)	0.0004	0.1101	0.9968
*S. sejunctum*	Dioecious		43				0.0000	0.1215	1.0000
		Gubara Pools	24	12.82928	132.8756	CTM 1739 (BUPL)	0.0000	0.1288	1.0000
		Barkk Sandstone	19	12.85907	132.81788	CTM 1729 (BUPL)	0.0000	0.0953	1.0000
*S. asymmetriphyllum*	Dioecious		49				0.0004	0.1582	0.9973
		Merl Rock	32	12.42622	132.96022	CTM 1702 (BUPL)	0.0001	0.1333	0.9993
		Bardedjilidji	16	12.43727	132.96803	CTM 1721 (BUPL)	0.0006	0.1825	0.9967
		Jabaluka	12	12.47702	132.90275	CTM 1700 (BUPL)	0.0011	0.1286	0.9920
*S. raphiotes*	Cosexual		83				0.0009	0.1778	0.9951
		Merl Rock	47	12.42622	132.96022	CTM 1709 (BUPL)	0.0004	0.1735	0.9977
		Bardedjilidji	24	12.43727	132.96803	CTM 1714 (BUPL)	0.0004	0.1439	0.9974
		Jabaluka	12	12.47702	132.90275	CTM 737 (CONN)	0.0038	0.1043	0.9635

## Data Availability

The data presented in this study are openly available on GitHub at https://github.com/cantley (accessed on 31 May 2023) and NCBI’s Sequence Read Archive (SRA) at https://www.ncbi.nlm.nih.gov/sra/docs/ (accessed on 31 May 2023).

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
