# Peer review of "A Foundational Population Genetics Investigation of the Sexual Systems of Solanum (Solanaceae) in the Australian Monsoon Tropics Suggests Dioecious Taxa May Benefit from Increased Genetic Admixture via Obligate Outcrossing"

_plants, 2023, doi:10.3390/plants12112200_

Round 1
Reviewer 1 Report
The article “A Foundational Population Genetics Investigation of the Sexual Systems of Solanum (Solanaceae) in the Australian Monsoon Tropics…” is devoted to the study of genetic diversity and structure of five Solanum species with different sexual systems and factors that can influence this. I think that the presented research is very well planned, conducted and presented in the manuscript. The article can be accepted for publication in present form.
I have only two small items to comment:
1. Page 4. Figure 1. Map indicating known collection localities of each of the six species for which population genetics data were analyzed in this study. Populations sampled in this study are indicated with white text. (a) View of the northwestern region of (b) Australia, … - It seems that A and B are confused.
2. Page 7, figure 2 and page 8, figure 3. – How can the authors explain, why the distance between S. ossicruentum and S. cowiei is so small (Fig. 3A), and pairwise Fst between them is more than 0.8 (Fig. 2A)?

Reviewer 2 Report
Dear authors, I appreciated your work as very good, scientifically significant, the MS is well written. I recommend its publication in Plants.
However, I have several editor’s comments. In particular, it seems that in the preliminary variant of MS six species were included, but later one was excluded, and not all corresponding changes were made.
1. The abstract (316 words) substantially exceeds the recommended length that should be a total of about 200 words maximum (Instructions for Authors)
2. According to the Materials five species represented “S. dioicum + S. echinatum Group” – (S. asymmetriphyllum, S. cowiei, S. ossicruentum, S. sejunctum and S. raphiotes) and were studied in the work, the corresponding data are given in most Tables and Figures. However, six species are mentioned in some places (Lines 124, 170, 192, 215, 233), though the sixth species S. ultraspinosum presents only on Figure 1. Isn’t this a mistake?
Line 171 “across 203 individuals representing 10 total populations”
However, in Table 1, the total number of individuals (n) is 193, besides ån for S. raphiotes is not given as for other species.
Line 233 “Figure 3. Multivariate analysis visualizations of all six species representing all 11 genetically determined populations of this study from 212 total individuals”
However, we see 5 ssp, 10 populations (and 193 individuals)?
3. Figure 3b. The population Merl Rock depicted by orange contains only 12 points whereas this is the largest sample (n=47). Isn’t this a mistake?
4. Line 561 GBS library preparation
Genotyping by Sequencing library preparation
5. In the text of the Results 2.4-2.5 (p. 8-10) latin names of the species are not in italics
6. It is necessary to check and correct References
Absent in the References:
Doyle & Doyle, 1987
Peterson et al (2012)
Absent in the text:
Anderson, G. J., Symon D. E. (1985)
Goldberg et al., 2017
Barrett R. L. (2013a) and Barrett R. L. (2013b) – letters after the year are absent in the References
Kamvar et al. 2015 – in the text
Kamvar Z. N., Tabima J. F., Grünwald N. J. (2014) – in the References
The commas after the last names are often absent
